# Flexible Context-Driven Sensory Processing in Dynamical Vision Models

**Lakshmi Narasimhan Govindarajan**[*1, 3, 4], **Abhiram Iyer**[*2, 3, 4],
**Valmiki Kothare**[1], and **Ila Fiete**[1, 3, 4]

[1]Department of Brain and Cognitive Sciences, MIT, Cambridge, MA
[2]Department of Electrical Engineering and Computer Science, MIT, Cambridge, MA
[3]McGovern Institute for Brain Research, MIT, Cambridge, MA
[4]K. Lisa Yang Integrative Computational Neuroscience (ICoN), MIT, Cambridge, MA ,
{lakshmin,abiyer,valmiki,fiete}@mit.edu

## Abstract

Visual representations become progressively more abstract along the cortical hierarchy. These abstract representations define notions like objects and shapes, but at the cost of spatial specificity. By contrast, low-level regions represent spatially local but simple input features. How do spatially non-specific representations of abstract concepts in high-level areas flexibly modulate the low-level sensory representations in appropriate ways to guide context-driven and goal-directed behaviors across a range of tasks? We build a biologically motivated and trainable neural network model of dynamics in the visual pathway, incorporating local, lateral, and feedforward synaptic connections, excitatory and inhibitory neurons, and long-range top-down inputs conceptualized as low-rank modulations of the input-driven sensory responses by high-level areas. We study this **D**ynamical **C**ortical **net**work (*DCnet*) in a visual cue-delay-search task and show that the model uses its own cue representations to adaptively modulate its perceptual responses to solve the task, outperforming state-of-the-art DNN vision and LLM models. The model's population states over time shed light on the nature of contextual modulatory dynamics, generating predictions for experiments. We fine-tune the same model on classic psychophysics attention tasks, and find that the model closely replicates known reaction time results. This work represents a promising new foundation for understanding and making predictions about perturbations to visual processing in the brain.

## 1   Introduction

We readily use abstract cues to modulate our sensory perception. These include cues to attend to abstract high-level features (find Waldo; count the number of hoop shots, etc.) or low-level features (find the red items, vertically oriented bars, etc.). Such cue-based modulations of the visual pathway allow us to locate items of interest more rapidly and accurately, and to perform goal-directed computations.

Understanding how top-down modulatory cues are represented and then interact with bottom-up sensory-driven neural responses has been a longstanding goal in computational cognitive neuroscience [1–4]. Extensive psychophysical experiments [5] and studies showing that individual neurons whose receptive fields are aligned with the attentional cue exhibit a gain modulation [6, 7] shed light

---

*Denotes equal contribution.

38th Conference on Neural Information Processing Systems (NeurIPS 2024).

on the phenomenon. However, a fundamental circuit-level and conceptual problem remains open: what is the nature of the "*modulatory homunculus*" that knows, given various goals and context cues, which low-level representations to modulate, in which combination and which topographic part of the representational space in different processing layers? We lack a cohesive computational framework to link these two levels of representation. Simultaneously, we lack models of sensory processing that fully take into account the recurrent and temporally unfolding nature of computation in the brain; thus, they fall short of explaining phenomena like reaction time variations with task difficulty and the sharpening of perception as information is integrated within a trial.

We combine known architectures of visual cortex with advances in machine learning to introduce a biophysically-inspired model, the **D**ynamical **C**ortical **net**work (*DCnet*), to solve cue-delay-visual search tasks (Figure. 1). The model is endowed with several relevant properties of biological circuits, including separate (tuned) excitatory and (weakly-tuned) inhibitory populations, lateral inhibition (intra-area recurrence), and neuron types with distinct learnable time constants. We operationalize the "modulatory homunculus" as multiplicative low-rank perturbations from higher-order cortical and thalamic areas. Specifically, these low-rank perturbations arise from the model's own sensory responses from earlier times within the trial.

**Contributions.** In this work, we focus on analyzing and interpreting the internal dynamics and behavioral modes of our biophysically inspired model on a suite of visual cue-mediated tasks.

- We introduce *vis-count*, a novel, challenging, and parametrically generated cue-delay-visual search task. A visual cue (consisting of a color, a shape, or a color-shape conjunction) specifies which objects in a subsequently presented scene of geometric objects to count.

- We present a biologically realistic model of the brain's visual system, the **D**ynamical **C**ortical **net**work (*DCnet*), that is relatively shallow with local and top-down feedback and separate E/I neurons, which is capable of top-down attentional modulation based on previously presented cues and processes information over time. Our framework is among the first to link levels of analyses from physiology to behavior via computations and is a necessary first step toward hypothesis generation for neuroscience.

- Our model outperforms state-of-the-art standard DNNs and LLMs on the task, while being interpretable and having orders of magnitude fewer parameters.

- We perform *in silico* electrophysiology of the circuit's population dynamics to show that cue-based modulation drives a divergence of the bottom-up responses from the uncued case.

- We can fine-tune the same model on new stimulus sets corresponding to classic human psychophysics attention tasks. Reaction time analogues in our model closely replicate experimental observations.

In sum, these contributions suggest that our approach is a promising framework for modeling the brain's visual processing dynamics, one that replicates many key attentional phenomena and that can generate testable hypotheses and predictions about circuit mechanisms.

## 2 Related Work

**Stimulus computable models of visual cognition.** Modeling top-down contextual effects on bottom-up sensory processing during visual search is of longstanding interest in the vision sciences community. There are a large number of phenomenological models of visual search and associated reaction time findings [8–12], but only recently have models been able to operate directly on high dimensional sensory inputs [13–16]. The most promising approach involves augmenting pre-attentive ventral stream models with controllers either via attention maps [13] or multiplicative modulation factors on network activities [17]. However, these models cannot be used to faithfully study sensory processing dynamics because they are purely feedforward. Recurrent models of early sensory processing with lateral feedback and distinct excitatory and inhibitory populations [18–20], and top-down feedback [21] have shown promise in accounting for contextual visual computations and human reaction times on visual cognitive tasks. However, to our knowledge, stimulus-computable recurrent vision models compatible with *cue-delay-target* paradigms do not exist. More generally, models have tended to either be stimulus-computable or grounded in biological realism, but usually not both.

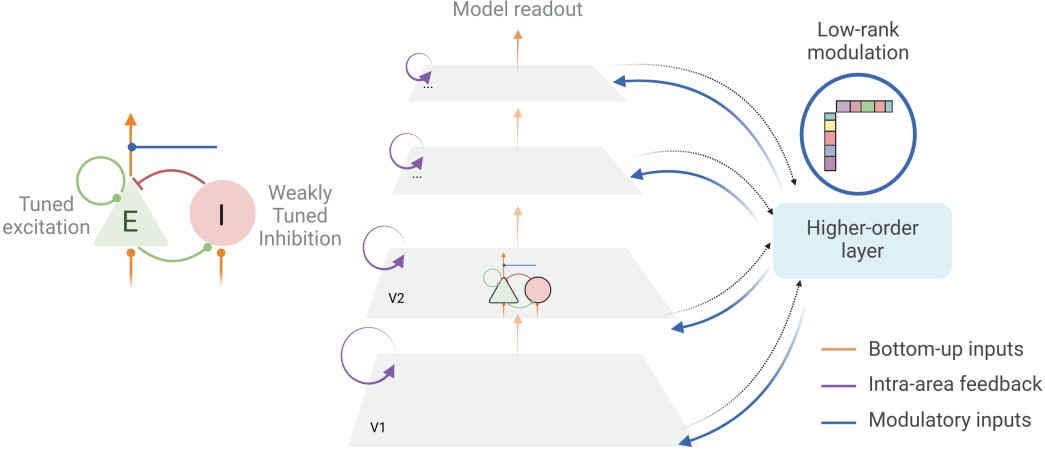

Figure 1: **Low-rank modulations to drive context-aware processing.** We present a biologically motivated, end-to-end trainable network model of dynamics in the visual pathway. Layers in the model are parameterized by recurrent (tuned) Excitatory (E) and (weakly tuned) Inhibitory (I) neural populations that interact bidirectionally with a higher-order layer in a low-rank manner. The low-rank modulatory factors serve to extract abstract context cues from sensory responses for subsequently driving neural dynamics into context-appropriate dynamical regimes.

**Low-rank interactions.** Neuronal networks in the brain can be involved in disparate computations simultaneously [22]. Computational neuroscience research has focused on understanding how such neural networks can represent task-specific information and, relatedly, its consequence on the system's overall behavior. Theoretical results suggest that the coexistence of structured low-rank connectivity and random connectivity in networks can enable multi-task computations and expand the dynamic range of the network's functional capability [23–27]. An alternate but non-orthogonal viewpoint is that low-rank control inputs from an external system can switch a network's input-output mappings context-dependently [28–30]. Most of the computational work in this regard is, however, rooted in the motor domain, with Schmid and Neumann being a recent exception.

**Neurophysiology of attentional control.** Biological attention upmodulates task-relevant information by filtering sensory responses using "templates". Empirical work shows that attentional templates are primarily represented in higher order cortical areas [31–34] and that such templates can be learned rapidly on a per-task basis [35]. Moreover, the nature of such attentional filtering is known to be cell-type and layer-specific [36], with theories emphasizing the role of top-down mediation of specific GABAergic interneurons [37]. In addition to cortical feedback inputs, higher-order thalamic nuclei are also known to convey contextual inputs to sensory regions either via direct projections [38, 39] or indirectly through the frontal cortices [40, 41]. Building biological sensory processing models that account for all these disparate findings is of primary importance to the Neuroscience community.

## 3 General methods

### 3.1 *DCnet* Model and training details

*DCnet* comprises two core components: a biologically motivated sensory perception stream and a higher-order area that interacts bidirectionally with it (Figure. 1). Each of the four sensory areas in our model are organized retinotopically as hypercolumns consisting of distinct excitatory and inhibitory neural subpopulations that obey Dale's law [42]. Excitatory pyramidal neurons receive bottom-up and recurrent lateral excitatory inputs as well as short-range lateral inhibitory inputs from interneurons in the same area. Interneurons receive bottom-up and lateral excitatory inputs. Finally, pyramidal neurons project in a feedforward manner to their downstream area. The ratio between excitatory and inhibitory neurons is $4 : 1$, as observed in cortical areas [43].

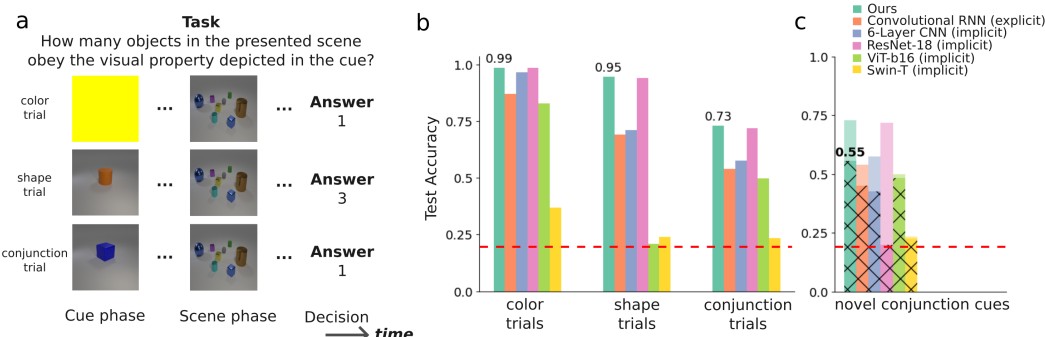

Figure 2: **Explicit context-guided modulation of sensory dynamics is necessary for learning generalizable solutions. a.** We introduce *vis-count*, a parametric, visually-cued, delayed search task. On each trial, models are cued with a visual attribute (either a color, shape, or a conjunction of the two), and after a delay, a scene is presented. The task is to count and report the number of cue-consistent objects in the scene. **b.** On each of the three types of trails, we find that our model consistently outperforms state-of-the-art standard DNNs (Section. 3.2; Baselines) on novel held-out scenes. *Implicit* models refer to the condition where cues and scenes are presented simultaneously. The red dashed line denotes chance performance. **c.** A harder test of generalization on novel scenes *and* cues reveal that our model is robust to such variations, unlike performant implicit models.

The higher-order layer receives pyramidal input from each area and, in turn, modulates inter-area projections in a low-rank manner operationalized as follows.

$$\mathbf{r}_1; \mathbf{r}_2 = \xi(\mathbf{h}_t^l)\mathbf{W}_{l,1}^T + \mathbf{b}_1; \xi(\mathbf{h}_t^l)\mathbf{W}_{l,2}^T + \mathbf{b}_2 \qquad \text{\# Compute the linear projections}$$

$$\mathbf{m}_t^l = \xi(\mathbf{h}_t^l)\left[\mathbf{r}_1 \otimes \mathbf{r}_2\right] \qquad \text{\# Compute the modulating factors}$$

$$\mathbf{h}_{t+\Delta}^l = \mathbf{h}_{t+\Delta-\epsilon}^l \odot \mathbf{m}_t^l \qquad \text{\# Execute the modulation}$$

Here, $\mathbf{h}_t^l \in \mathrm{R}^{C \times H \times W}$ is the excitatory population readout of area $l \in [1..4]$ at time $t$; $\xi(.)$ is the spatial average pooling operator; $\otimes$ denotes outer product and $\odot$ denotes pointwise scaling.

Neurons have learnable cell-type specific time constants. We dispense of traditional ML operations such as BatchNorm or LayerNorm, designed to impart stability during training. Our model has $\sim 1.8$M parameters that we learn via gradient descent on a task-performance objective. A full mathematical specification of the model is provided in Appendix. A.1.

### 3.2 Baselines

We instantiate baseline models of two varieties. First, we consider a "Convolutional RNN" model ($\sim 8.8$M params) that uses a traditional 6-Layer convolutional backbone feeding into a gated recurrent unit (GRU) [44] with $N = 2048$ neurons (Appendix. B). We construct this baseline to evaluate the benefits of an explicit modulatory mechanism such as the higher-order layer in our model. Second, we consider four standard deep feedforward neural network architectures. As these models do not operate on spatiotemporal inputs, we condense trials to a single time point by stacking cues and scenes together. Given the lack of an explicit cue followed by scene phase for these models, we term them *implicit*. Cues and scenes are presented at the same time. These baselines provide an upper bound on the expected performance of *DCnet* and verify that our chosen task is a non-trivial computational challenge. In our experiments, we consider the following implicit models: a 6-Layer CNN ($\sim 2.8$M params); ResNet-18 [45] ($\sim 11.5$M params); ViT-B/16 [46] ($\sim 86$M params), a vision transformer with a patch size of 16px; Swin-T [47] ($\sim 28$M params), a hierarchical vision transformer. Furthermore, we also ran experiments on the zero-shot generalization abilities of an LLM (Appendix. C).

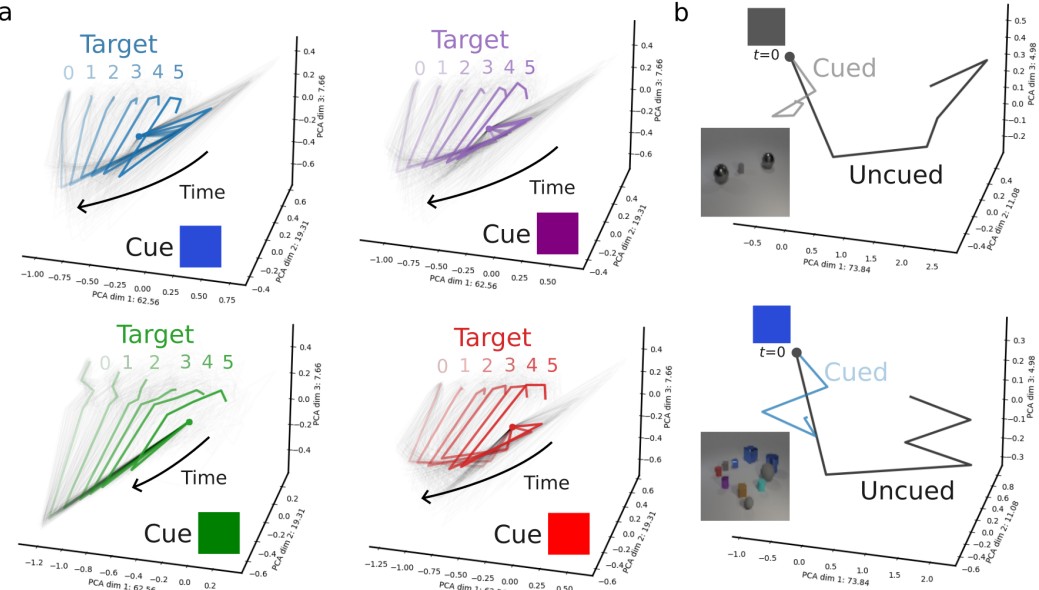

Figure 3: **Neural network dynamics reveal context-dependent behavior.** We perform dimensionality reduction on late layer model activities and visualize neural trajectories (**Gray** for individual trials; Dark-colored for trial averages) for different experimental manipulations. The solid dot indicates the trial start. **a.** For a fixed cue (color trials visualized here), network dynamics reflect the extraction and preservation of task-relevant information (object counts) while being invariant to task-irrelevant bottom-up responses from the different scenes. **b.** Matched cued vs. uncued trials for the same scene (inset) reveal a divergence of the bottom-up responses driven by the low-rank modulations.

## 4   vis-count: A parametric cued visual search task

### 4.1   Task and stimuli

We draw inspiration from Clevr [48], a synthetic dataset for language-mediated visual reasoning and construct *vis-count*, a parametric visually-cued, delayed search task. On each trial, we challenge models to count and report the number of geometric objects in a visual scene with features consistent with a cue provided earlier (Figure. 2a). Cues can be simple colors, geometric shapes, or a conjunction of the two. Scenes comprised $3-10$ geometric 3D shapes of varying sizes, colors, and material properties. By construction, there can be $0-5$ cue-consistent objects in the scene. We counterbalanced the dataset so that the distribution of target counts was uniform across trial types. Cues and scenes were rendered at $320 \times 240$px and resized to $128 \times 128$px for model training and evaluation. In total, our training (validation) dataset comprised of $\sim 384$K ($38$K) trials. As with Clevr, we detect and discard scenes with fully occluded objects.

For *DCnet* and the Convolutional RNN baseline, cues and scenes are visual inputs provided to Layer 1 of these models. Cues are persistently provided to the network for the first $T$ discrete time steps followed by scenes for the next $T$ time steps. The output activities of the last layer at the last time step ($t = 2T$) are transformed into logits, and a supervision signal is provided via a cross-entropy loss, with ground truth labels counted from 0 to 5. For the "implicit" baseline models, cues and scenes are *stacked* together and presented simultaneously.

### 4.2   Results

We report the results of our model evaluations and comparisons to baselines in Figure. 2b-c. Our model consistently and significantly outperforms state-of-the-art DNN vision models when evaluated on trials with novel held-out scenes, achieving $99\%$, $95\%$, $73\%$ accuracy on the color, shape, and conjunction trials, respectively. Additionally, we perform a harder generalization test by synthesizing trials with novel cues *and* novel scenes (e.g., we test our model on blue and green colors, cube

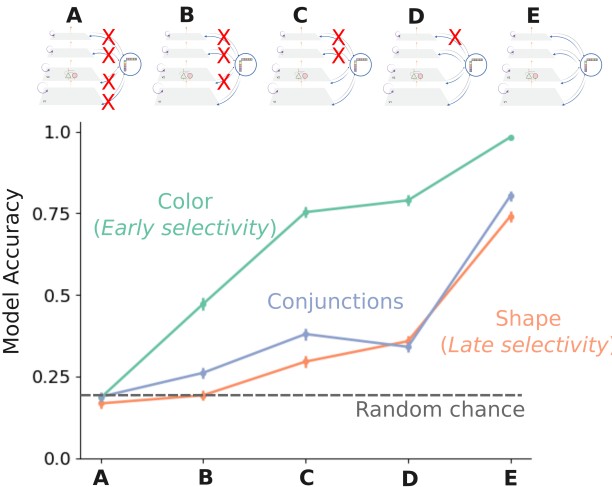

Figure 4: **A selectivity gradient emerges across the cortical layers for the different visual cues.** We sequentially lesion the modulatory synapses in a trained *DCnet* from the higher-order layer to each sensory area, starting from the top-most (D) to the bottom-most (A) sensory area, while documenting task performance on each trial type. (E) indicates the intact *DCnet*. While late lesions have a minimal effect on performance in color trials, the opposite is true for shape trials. Early areas in the *DCnet* exhibit color selectivity, while later areas exhibit shape selectivity.

shapes, and conjunctions thereof, all unseen in the training set). Our model demonstrated the best generalization performance (55%) while even the most performant baseline models (with orders of magnitude more parameters) dropped to chance (16.67%). Sample model results are visualized in Appendix. A.5.

Model errors, too, can be particularly enlightening. Consider the last example in Figure. A.5. When cued to find "blue cylinders", the model mistakenly appears to include either the cyan cylinder or the occluded blue cube in its count, either of which would reflect an *illusory conjunction* [49]. Illusory conjunctions arise due to failures of feature-based attention and have been extensively studied in the human cognition literature. The presence of such pathologies in our model exposes an opportunity to gain potential insights into the neural implementations of feature-based attention.

In the following sections, we report analyses of our model's learned dynamics and internal representations that support contextual guidance.

***What to modulate?* Low-rank structures support optimal cue representations.** In daily life, "cues" are not specified to us as abstract rules but rather as high-dimensional sensory inputs. How do we construct a representational space where abstract rules (derived from sensory responses) and the sensory responses themselves are distinct yet coexist? Following insights from electrophysiology [29] and prior theoretical work [23, 24, 28], we hypothesized that learning to inject derived cue information back into sensory representations as low-rank perturbations will aid in optimally partitioning the sensory representational space.

*DCnet*'s task performance indicated that it had indeed learned task-optimal representations. Here, we perform dimensionality reduction on the activities of pyramidal neurons in the last layer of *DCnet* to probe how this optimality emerges.

First, we observe that cued vs. uncued dynamics on individual trials become progressively divergent and nearly orthogonal with time (Figure 3b). This indicates that the bidirectional interactions likely promote the formation of low-dimensional activity subspaces embedded within the higher-dimensional ambient activity space. Second, we see that the bottom-up responses to the same set of scenes are differentially modulated to appropriate target subspaces based on the cue (Figure 3a). We believe that this happens through the inactivation of context-irrelevant subspaces, resulting in invariance to current task-irrelevant features. As additional consideration, we note that training *DCnet* without this top-down feedback mechanism brings performance down to 38%, highlighting the importance of this interaction in our framework.

***Where to modulate?* A cortical gradient for feature selectivity.** After extracting abstract context rules from sensory responses, a question remains: At what level of the representational hierarchy must the perturbation be applied? We perform lesion analysis on the trained *DCnet* to probe this question (Figure. 4).

We sequentially "turn off" modulations, area by area, and observe their detrimental effects on overall function as determined by task performance. We implement this by setting the modulating factors to

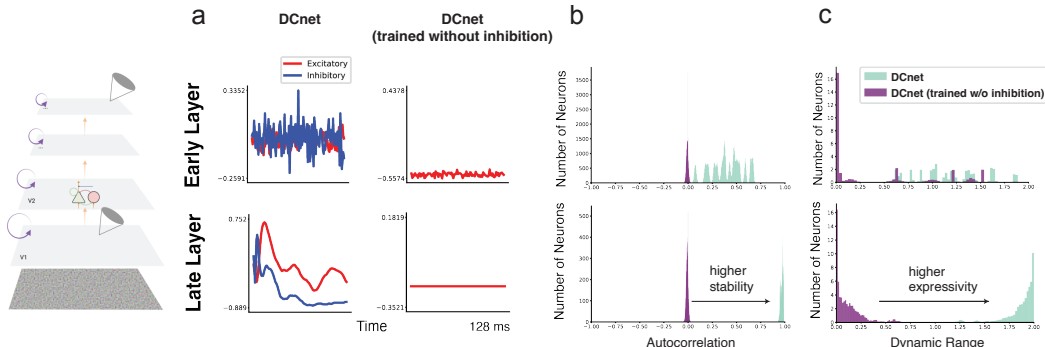

Figure 5: *In silico* **electrophysiology sheds light on tuning properties and excitation-inhibition dynamics.** We drive network activity in *DCnet* with uncorrelated, time-varying Gaussian noise inputs. (a) We depict a randomly chosen neuron in early and late sensory areas of *DCnet*. Each E/I neuron pair shown here has matched receptive fields. Learned time constants reflect fast/slow integration in early/late areas, revealing a macroscopic gradient. By contrasting *DCnet* trained with and without inhibition, we find that inhibition plays a crucial role in (b) imparting stability and (c) expanding the dynamic range of computations in the network. More quantitative details are presented in Appendix. A.4.

**1**. We hypothesized that if the modulation of pyramidal neuron activity in a given area is essential for the system's function, then lesioning this perturbation will result in a maximal drop in performance. Interestingly, this analysis yielded differential results per trial type (Figure. 4). For the color trials, we find that lesioning modulations in the early areas had the largest impact on performance, while for the shape trials, it was the late modulations that proved critical. In contrast, modulation strength seemed relatively uniform across the sensory areas.

Taken together, these results offer two insights. First, opposing cortical gradients for color and shape selectivity emerge and coexist in the sensory areas despite the model only being supervised to "count". Second, leveraging this selectivity gradient, the higher-order area learns to apply appropriate low-rank modulations. We believe that the low-rank nature of the context-based modulation, not only within but also across areas, is fundamental to generalization.

**Excitation-Inhibition Dynamics.** The amplification of context-relevant sensory representations must be balanced to support stable dynamics [50]. The neural underpinnings of this balance and its computational role in feature gain modulation have previously only been studied phenomenologically. Here, we leverage *DCnet* to investigate how the circuit-level properties discovered through optimization can support overall network function.

We probe the cell-type specific neural dynamics in *DCnet* by driving network activity with uncorrelated, time-varying Gaussian noise inputs. First, despite fewer interneurons in the model, inhibitory interactions play a crucial role in imparting stability (Figure. 5b) and expanding the dynamic range of computations expressed by the network when compared to a version of *DCnet* trained without inhibition (Figure. 5c). Second, we detect the presence of co-tuned excitation and inhibition (Appendix. A.4). Empirically, co-tuning is known to be a common organizational principle across the sensory areas. These findings imply the critical role of interneurons in cortical amplification dynamics underlying context-dependent computations. Finally, we observe that neurons learn to integrate information faster in early vs. late areas, revealing the emergence of a macroscopic gradient in neural timescales (Figure. 5a, 8).

## 5   Model psychophysics on parametric cued feature searches

A feature search is a variant of the general visual search problem in which a "target" is defined by a single discriminative feature [5]. Parametric variations along (or orthogonal to) this discriminative feature axis help shed light on the mechanisms of top-down contextual guidance and its impact on behavior. Here, we consider two feature search tasks from the human psychophysics literature.

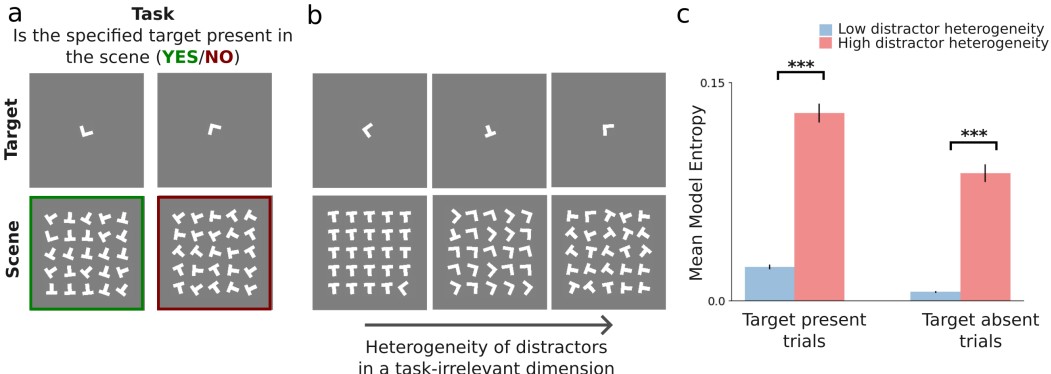

Figure 6: **Slower reaction times/larger output entropy with more heterogeneous distractors. a.** Task description. **b.** We parametrically vary the heterogeneity in distractor orientation (similar to [5]). **c.** In both target-present and target-absent trials, we find that the *DCnet*'s output entropy is significantly different between low distractor heterogeneity vs. high distractor heterogeneity trials. Error bars represent the standard error of the mean.

**General methods.** To perform model psychophysics, we start from the *DCnet* model trained on *vis-count*. We replace its readout to do binary classification (target present/absent) and fine-tune the model on each task below. Furthermore, to derive a measure of model "reaction time," we compute the entropy of *DCnet*'s decision outputs after evolving dynamics for the duration of each trial.

### 5.1 The role of distractor heterogeneity

**Task and stimuli.** A target feature's bottom-up salience (pop-out) does not survive variations in irrelevant distractor features. (Figure. 6b) [5]. To test this, we construct cued-feature search trails where models search for an L/T (target) in a grid with a fixed number of Ts/Ls respectively (scene). Targets and scenes were rendered on a $128 \times 128$px canvas. Targets (L/T) were presented at the center and oriented uniformly at random between $[0, 2\pi)$. The distractor heterogeneity level ($\omega$) for every trial was uniform random between $[0, 1]$. Consequently, distractor orientations in the scene were uniform random between $[\theta - \omega\frac{\pi}{2}, \theta + \omega\frac{\pi}{2}]$ with $\theta \in [0, 2\pi)$ being the mean distractor orientation. The target was present in $50\%$ of the trails. Our training (test) dataset comprised of 32K (8K) trials.

**Results.** When fine tuned on this task, *DCnet* achieved an overall accuracy of $97\%$ on the test trials. Moreover, faithful to the human psychophysics results, we find that the entropy of *DCnet* outputs is significantly different for low ($\omega <= 0.5$) vs. high ($\omega > 0.5$) distractor heterogeneity for both target-present (Mann–Whitney $U = 302947., p < .001$) and target-absent trials (Mann–Whitney $U = 301376., p < .001$). While target-absent trials yield higher reaction times in humans when compared to target-present trials, we don't see this in our model's output entropy. This possibly reflects an imperfect choice for a model reaction time measure, an aspect of our work that we look to extend upon in future work.

### 5.2 The role of target-distractor feature differences in the presence of distractors

**Task and stimuli.** In a classic critical color difference task, human participants were required to detect a target that differed from distracting stimuli only in color [51]. Search times were measured for varying color differences as a function of display density (the number of distractors). We parametrically synthesize stimuli to recreate this task (Figure. 7a-b).

Target and distractor stimuli were chosen to be circular discs of radii 10px. Cues were rendered at the center of a $128 \times 128$px canvas at a target color chosen at random from among four perceptually uniform color spaces. Search scenes consisted of $1 - 7$ distractor stimuli whose color differed from the target color at one of 10 preset target-distractor difference levels chosen uniformly at random. The target was present in $50\%$ of the trials. In total, our training (test) dataset comprised of 32K (8K) trials.

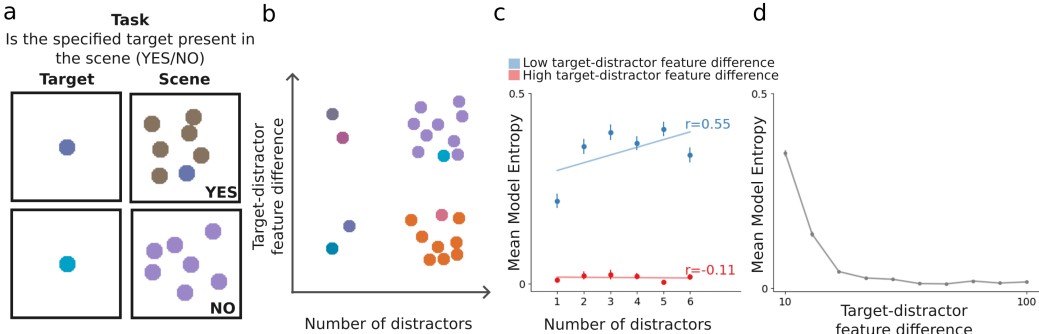

Figure 7: **Differential reaction times/output entropy predictions for varying levels of target-distractor differences. a.** Task description. **b.** We parametrically vary both the number of distracting stimuli in the scene and the target-distractor color difference [5]). **c.** *DCnet*'s output entropy is invariant (linear) to the number of distracting stimuli when the feature difference is high (low), capturing the psychophysical phenomenon of "popout." **d.** Model entropy as a function of target-distractor feature difference when marginalized over the number of distracting stimuli. Error bars represent the standard error of the mean.

**Results.** When fine tuned on this task, *DCnet* achieved an overall accuracy of $95\%$ on the test trials. Moreover, *DCnet* recapitulates two key findings from the psychophysics literature.

We probe *DCnet*'s output entropy (a proxy for model RT) as a function of the number of distractors in the scene. We find that for trials in which the target-distractor color difference was high, model entropy was invariant to the number of distracting stimuli (Pearson's $r = -0.11$). On trials where the target-distractor color difference was low, model entropy increased linearly with the number of distracting stimuli in the scene (Pearson's $r = 0.55$). We find a similar result when we probe *DCnet*'s output entropy as a function of the target-distractor feature difference. These results faithfully replicate behavioral effects reported in prior literature [5, 51, 52].

As we alluded to in Section 5.1, our reaction time (RT) metric is one among many possible choices. To take a step further, we implement and test another RT metric inspired by evidential learning (EDL) theory as proposed in [20]. We verify that our primary conclusions from this experiment hold even in the context of this new metric. Finetuned on the EDL objective, *DCnet* achieves $86\%$ accuracy on this task. Model RTs, computed as a time-averaged uncertainty measure, increased linearly with the number of distractors in the low target-distractor color difference trials (Pearson's $r = 0.92$) and was invariant to the number of distractors on high target-distractor color difference trials (Pearson's $r = 0.2$).

## 6 Discussion

The advent of highly-performant ventral stream models of visual perception is rapidly revolutionizing visual neuroscience research. Stimulus computable models operating directly on high dimensional sensory inputs yield benefits as hypothesis generators for scientific discovery. [53–55]. However, there exist two fundamental axes of dissonance, which are potential rate limitors. First, the emphasis on building "bigger and better" models promotes a divergence from biological realism [56]. Second, it is evident that the extent of biological visual capabilities spans a space bigger than one that entails only feedforward perceptual modules [1, 57, 58]. It calls for accounting for the cooperative dynamics between perceptual and cognitive processes [59].

In this work, we aim to bridge this gap by introducing a computational framework that emphasizes biological realism and conceptualizes interactions between perceptual dynamics and abstract cognitive demands while being scalable and stimulus-computable. We present the **D**ynamical **C**ortical **net**work (*DCnet*): a trainable neural network model of visual dynamics incorporating local, lateral, and feedforward synaptic connections, excitatory and inhibitory neurons, and long-range top-down inputs conceptualized as low-rank modulations of the input-driven sensory responses by high-level areas.

We start by studying the ability and behavior of *DCnet* to operate in a visually-cued search paradigm. *DCnet* not only outperforms state-of-the-art DNN models, but its population states over time reveal the computational structure and neural underpinnings of contextual modulatory dynamics, generating predictions for experiments. Furthermore, we fine-tune the same model to perform two classic 2AFC attention psychophysics tasks. Reaction time analogs from *DCnet* strikingly recapitulate core tenets of feature-based attentional modulation. We find that *DCnet* responses are sensitive to target-distractor feature differences, heterogeneity of irrelevant distractor features, and display density.

Overall, these contributions suggest that our approach is a promising framework for modeling the brain's visual cortical dynamics, one that replicates key neural and behavioral signatures of contextual attentional modulation.

**Limitations and future directions.** In this work, we take the first step towards building an overcomplete model of cortical circuitry. Palpable omissions from our current framework include long-range inter-area feedback and compartmental separation of feedforward and feedback inputs. We hope to continue building on this framework in the future to include these components that will open up novel avenues for computational neuroscience. Additionally, extracting reaction time measures from large-scale recurrent models is a discipline of its own [20]. We adopt a simple reaction time metric here that suffices for our current purpose, but we plan to include other sophisticated comparisons in future work. Lastly, we have restricted our purview to visual sensory processing. The concept of contextual modulation, however, is pervasive across sensory modalities. We are excited about extending our ideas to other sensory domains.

**Broader impact.** Artificially intelligent models with enhanced visual capabilities are now pervasive in our daily lives. The not-so-hidden cost we pay to enjoy these models' benefits is their carbon footprint on our environment. Building energy-, parameter-, and sample-efficient models that are also performant is non-negotiable going forward. Understanding context-aware behavior is of utmost importance for neuroscience research as its failure modes are associated with several psychiatric and neuropathologies. We do not anticipate any negative impact that our work would create.

## Acknowledgments and Disclosure of Funding

We thank the McGovern Institute and the K. Lisa Yang Integrative Computational Neuroscience (ICoN) Center for supporting and funding this research. We are grateful to Mark Harnett, Jim DiCarlo, and Bob Desimone for enlightening conversations.

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

# A  Dynamical Cortical network (*DCnet*)

## A.1  Architectural specification

We instantiate *DCnet* as a recurrent convolutional neural network model following the primate visual cortex's anatomical and neurophysiological properties. *DCnet* implements local, lateral, and feedforward synaptic connections, excitatory and inhibitory neurons, and long-range top-down inputs conceptualized as low-rank modulations of the input-driven sensory responses by high-level areas.

Specified below are the governing dynamics for the neurons in our model. Neurons in our model are either excitatory (E) or inhibitory (I). $l$ denotes the network layer (brain area). $\alpha^l$ and $\beta^l$ are indices used to describe excitatory and inhibitory cell types, respectively, in layer $l$. $(x, y)$ describes a specific spatial location. $z^{(l)}$ is the feedforward input to layer $l$. $h$ refers to a neuron's instantaneous state.

$$\tau_{\mathrm{E}_{\alpha}l} \frac{dh_{\mathrm{E}_{\alpha}l}^{(l,x,y)}}{dt} = -h_{\mathrm{E}_{\alpha}l}^{(l,x,y)} + g_{\mathrm{E}}(z^{(l)}, h_{\mathrm{E}}^{(l)}, h_{\mathrm{I}}^{(l)}, \alpha^l, x, y)$$

$$g_{\mathrm{E}}(z^{(l)}, h_{\mathrm{E}}^{(l)}, h_{\mathrm{I}}^{(l)}, \alpha^l, x, y) = f([W_{\mathrm{input}\to\mathrm{E}}^{(l)} z^{(l)} + \lfloor W_{\mathrm{E}\to\mathrm{E}}^{(l)} \rfloor_{+} h_{\mathrm{E}}^{(l)} + \lfloor W_{\mathrm{I}\to\mathrm{E}}^{(l)} \rfloor_{-} h_{\mathrm{I}}^{(l)}]_{\alpha^l, x, y} + b_{\mathrm{E}_{\alpha}l})$$

$$\tau_{\mathrm{I}_{\beta}l} \frac{dh_{\mathrm{I}_{\beta}l}^{(l,x,y)}}{dt} = -h_{\mathrm{I}_{\beta}l}^{(l,x,y)} + g_{\mathrm{I}}(z^{(l)}, h_{\mathrm{E}}^{(l)}, h_{\mathrm{I}}^{(l)}, \beta^l, x, y)$$

$$g_{\mathrm{I}}(z^{(l)}, h_{\mathrm{E}}^{(l)}, h_{\mathrm{I}}^{(l)}, \beta^l, x, y) = f([W_{\mathrm{input}\to\mathrm{I}}^{(l)} z^{(l)} + \lfloor W_{\mathrm{E}\to\mathrm{I}}^{(l)} \rfloor_{+} h_{\mathrm{E}}^{(l)}]_{\beta^l, x, y} + b_{\mathrm{I}_{\beta}l})$$

$\tau_{\mathrm{E}_{\alpha}l}$ and $\tau_{\mathrm{I}_{\beta}l}$ are cell-type specific learnable neural time constants. Synaptic connections $\boldsymbol{W}$'s are sparse matrices on which we impose translational invariance (details below). These are, in practice, realized as convolutions. $b_{\mathrm{E}_{\alpha}l}$ and $b_{\mathrm{I}_{\beta}l}$ are excitatory and inhibitory cell-type specific learnable thresholds. $f(.)$ is a non-linear activation function. We use the hyperbolic tangent as our activation function $f$. An average pooling operation `Pool` is applied to layer pyramidal outputs ($h_{\mathrm{E}}$) to increase the receptive field size by a factor of two.

$$z^{(l+1)}[t] = \begin{cases} \mathtt{Pool}(h_{\mathrm{E}}^{(l)}[t]) \odot \Gamma(\xi(h_{\mathrm{E}}^{(l)}[t-T])) & t \geq T \\ \mathtt{Pool}(h_{\mathrm{E}}^{(l)}[t]) & t < T \end{cases}$$

We make discrete time approximations to train our model. The cue is presented first to the network ($z^{(1)}[0]$)) and the dynamics are unrolled for $T$ steps followed by scene presentation ($z^{(1)}[T]$)) for another $T$ steps. While the scene is presented, inputs to each layer are modulated as follows, where $\xi(.)$ is a pooling operator that computes the average activity per cell type across all $(x, y)$. $\Gamma(\mathbf{e})$ is the low-rank modulation function defined as follows:

$$\Gamma(\mathbf{e}) = \mathbf{e} \odot \sigma\left([\mathbf{W}_{l,1}\mathbf{e}^T + \mathbf{b}_1] \otimes [\mathbf{W}_{l,2}\mathbf{e}^T + \mathbf{b}_2]\right)$$

Here, $\mathbf{W}_{l,1}, \mathbf{W}_{l,2}$ are learnable linear projections and $\mathbf{b}_1, \mathbf{b}_2$ are learnable biases. $\otimes$ denotes outer product and $\odot$ denotes pointwise scaling. By construction, the output of $[\mathbf{W}_{l,1}\mathbf{e}^T + \mathbf{b}_1] \otimes [\mathbf{W}_{l,2}\mathbf{e}^T + \mathbf{b}_2]$ is a low-rank matrix.

## A.2  Implementational details

This section provides specific details on model parameters detailed in Section A.1. All layers have the following convolutional kernels ($\mathbf{W}$s specified in the governing equations): Input to excitation ($W_{\mathrm{input}\to\mathrm{E}}$), excitation to excitation ($W_{\mathrm{E}\to\mathrm{E}}$), excitation to inhibition ($W_{\mathrm{E}\to\mathrm{I}}$), inhibition to excitation ($W_{\mathrm{I}\to\mathrm{E}}$), and input to inhibition ($W_{\mathrm{input}\to\mathrm{I}}$). The dimensionality of these convolutions is listed layer-wise below, along with the kernel and padding shapes for all convolutions in that layer.

|         | Input size | # exc. cell types ($N_E^l$) | # inh. cell types ($N_I^l$) | Kernel size, Padding |
|---------|------------|------------------------------|------------------------------|----------------------|
| Layer 1 | $3 \times 128 \times 128$ | 16 | 4 | $(5, 5), (2, 2)$ |
| Layer 2 | $16 \times 64 \times 64$ | 32 | 8 | $(5, 5), (2, 2)$ |
| Layer 3 | $33 \times 32 \times 32$ | 64 | 16 | $(5, 5), (2, 2)$ |
| Layer 4 | $128 \times 16 \times 16$ | 128 | 32 | $(3, 3), (1, 1)$ |

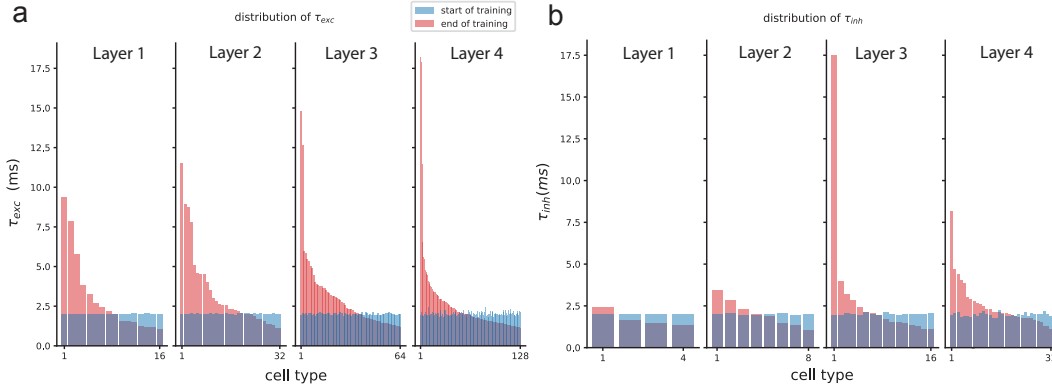

Figure 8: **An emergent macroscopic gradient in neuron time constants.** Cell-type specific integration time constants in *DCnet* are learnable parameters. While we initialize the $\tau$'s uniformly (blue) pre-training, we observe layer-dependence post-training for both the (a) excitatory, and (b) inhibitory cell types.

The outputs from the last layer are transformed via a linear readout parameterized as a fully connected layer $(1024 \times 6)$ into logits, which we supervise with a CrossEntropy loss. The model does not include standard embellishments like explicit normalization layers, model ensembling, test-time augmentations, etc.

### A.3    Training details

All models were trained on A100 GPUs for 100 epochs each. An experimental run took anywhere from 4-8 hours to complete. We used an AdamW optimizer (momentum=0.9, $\beta_1 = 0.9$, $\beta_2 = 0.999$), a one-cycle learning rate scheduler with a warm-up period of 30 epochs and a maximum learning rate of $4e - 4$. *DCnet* was 4 layers deep ($\sim 1.8$M learnable parameters) and was trained with batches of 256 samples. Code and datasets can be found here: Project repository.

### A.4    Mechanistic interpretability analyses

In the interest of mechanistic interpretability, we perform a series of experiments on *DCnet*. We drive activity by uncorrelated, time-varying Gaussian noise inputs. We then compute:

1. The Lag-1 autocorrelation as a measure of **stability** of the excitatory neurons (Fig. 5b). We find that *DCnet* excitatory neurons are significantly more stable than excitatory neurons in the *DCnet* (trained w/o inhibition) model. A Kolmogorov–Smirnov test confirmed the significant difference (statistic=1.0, $p < .001$).

2. The **Dynamic Range (DR)** of excitatory cells in each layer of *DCnet* and compare that to the DR of corresponding excitatory cells from *DCnet* (trained w/o inhibition) (Fig. 5c). DR is computed as the Interquartile range (a measure of statistical dispersion) of a neuron's activity over a time period of $128ms$ when driven by noise. We take the mean over 64 trials for each neuron and plot this distribution per layer in Fig. 5c. We find that excitatory neurons in the *DCnet* model have a significantly higher DR across layers compared to *DCnet* (trained w/o inhibition), suggesting the role of inhibitory interactions in expanding the range of computations carried out by each neuron. A Kolmogorov–Smirnov test confirmed these significant differences (Layer 1 (statistic=0.667, $p < .001$); Layer 2 (statistic=0.99, $p < .001$); Layer 3 (statistic=1.0, $p < .001$); Layer 4 (statistic=0.97, $p < .001$)).

3. The E-I correlation coefficient as a measure of **co-tuning** in *DCnet*. We find that the average (across neurons) E-I correlation is as follows: $-0.076$ (Layer 1), $0.766$ (Layer 2), $0.699$ (Layer 3), $0.535$ (Layer 4).

Additionally, we also highlight a macroscopic gradient in learned time constants across the layers of *DCnet* (Fig. 8).

## A.5 Sample model outputs

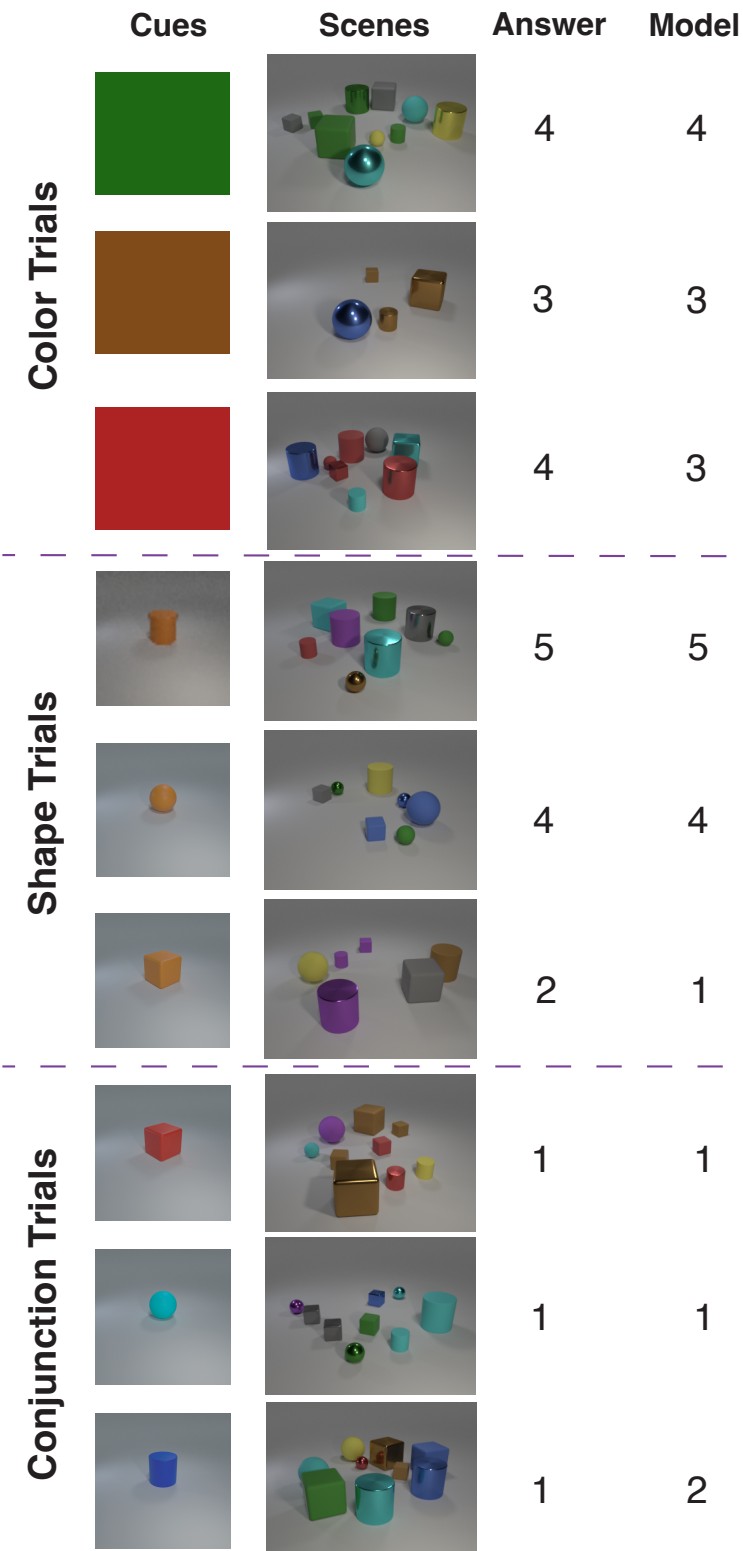

Figure 9: **Model outputs.** Visualizing example *DCnet* predictions on out-of-distribution (held-out) scenes across trial types.

### A.6 *DCnet* on visual object recognition

In a follow-up experiment, we train the sensory backbone of *DCnet* on a visual object recognition task. On CIFAR-10, we report a test accuracy of $84.79\%$. We highlight that our model does not include standard embellishments used in machine learning, including explicit normalization layers, model ensembling, test-time augmentations, etc. This is a first step that strongly demonstrates the potential of our framework to scale up.

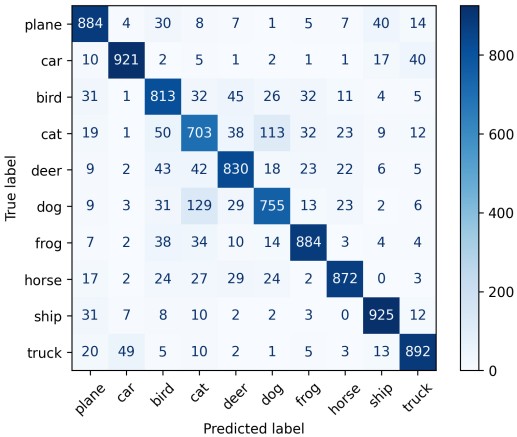

Figure 10: *DCnet* **performance on visual object recognition.** Confusion matrix of *DCnet* outputs on the CIFAR-10 test set. Overall model performance is $84.79\%$.

## B  Convolutional RNN Baseline

| Layer | Input Size | Operations |
|---|---|---|
| Layer 1 | $3 \times 128 \times 128$ | `Conv2d` ($8 \times 3 \times 5 \times 5$), `ReLU`, `AvgPool2d` ($5 \times 5$, stride 2) |
| Layer 2 | $8 \times 64 \times 64$ | `Conv2d` ($16 \times 8 \times 5 \times 5$), `ReLU`, `AvgPool2d` ($5 \times 5$, stride 2) |
| Layer 3 | $16 \times 32 \times 32$ | `Conv2d` ($32 \times 16 \times 5 \times 5$), `ReLU`, `AvgPool2d` ($3 \times 3$, stride 2) |
| Layer 4 | $32 \times 16 \times 16$ | `Conv2d` ($64 \times 32 \times 3 \times 3$), `ReLU`, `AvgPool2d` ($3 \times 3$, stride 2) |
| Layer 5 | $64 \times 8 \times 8$ | `Conv2d` ($128 \times 64 \times 3 \times 3$), `ReLU`, `AvgPool2d` ($2 \times 2$, stride 2) |
| Layer 6 | $128 \times 4 \times 4$ | `Conv2d` ($128 \times 128 \times 3 \times 3$), `ReLU`, `AvgPool2d` ($1 \times 1$, stride 1) |
| Projection | \multicolumn | `Fully connected` (2048 inputs, 2048 outputs), `ReLU`, `GRU` (2048 inputs, 2048 outputs) |
| Readout | \multicolumn | `Fully connected` (2048 inputs, 6 outputs) |

Table 1: Baseline model details

Convolution filters are mentioned as: (output channels $\times$ input channels $\times$ kernel height $\times$ kernel width). Pooling layers are mentioned as: (kernel height $\times$ kernel width, stride). Inputs to every layer were normalized with LayerNorm. The convolutional backbone serves as a feature extractor for the GRU network. We train this baseline model by passing in the cue for the first $T$ time steps followed by the scene for the next $T$ timesteps. For our experiments, we set $T = 3$. A cross-entropy loss is used for the readout activities at the final time step.

# C    Zero-shot testing of LLM

**Prompt**  Here is a task that takes two images, a cue and a scene. Here are the instructions:
1. If the cue is a color, return a count of how many objects in the scene have that color. 2. If the cue is an orange colored shape, return a count of all objects in the scene that have that shape. 3. If the cue is a non-orange colored shape, return a count of all objects in the scene that have that shape and color. You will only need to follow 1 of the 3 instructions above. Don't write code. There is no need to explain the logic of your counting. Simply provide the final answer.

**GPT4**  Sure, I can help with that. Please upload the two images you have: the cue image and the scene image, and I'll proceed according to your instructions.

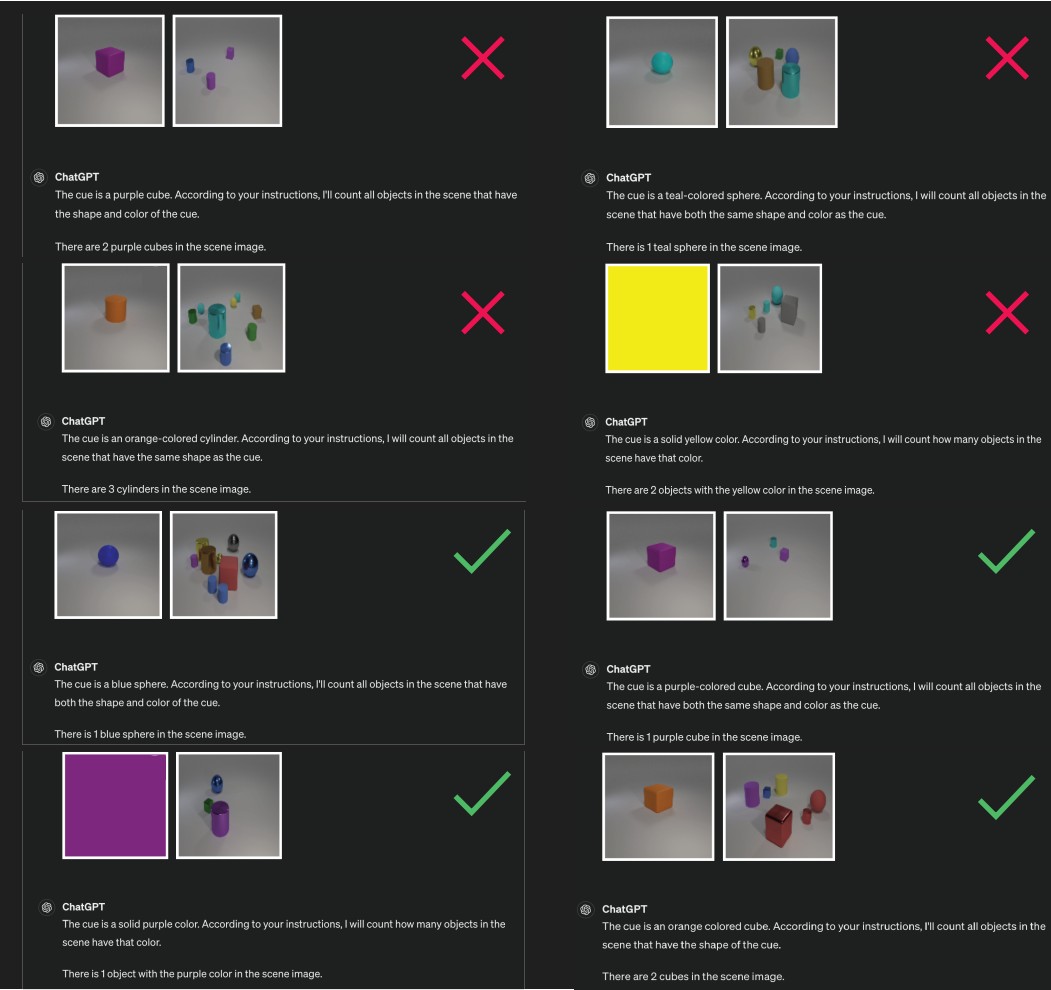

Figure 11:  **ChatGPT4 experiments** We prompted GPT4 to solve solve *vis-count*. Given a color, shape, or conjunction cue, we asked the LLM to identify how many objects in the provided scene match the cue's properties. Out of 30 trials, GPT4 achieved a success rate of 37%.

