# OpenReview forum: "Flexible Context-Driven Sensory Processing in Dynamical Vision Models"
_NeurIPS.cc/2024/Conference — NeurIPS 2024 poster_

### Official Review · Reviewer_BdqG · 2024-07-10

**Soundness:** 3
**Presentation:** 2
**Contribution:** 3
**Rating:** 6
**Confidence:** 3

**Summary:**

Artificial neural networks are loosely based on biological neural networks. In this study, the authors construct neural networks whose structures are based on general principles of visual signal pathway such as retinotopy. They used this model to ask if the context-dependent feedback mimicking top-down signals in the brain could help artificial neural networks perform cue-delay tasks more efficiently.

**Strengths:**

The authors constructed a simple but effective model (DCNet) that can capture the gist of interplay between low-order sensory areas and high-level cognitive areas. They showed that their model can outperform conventional deep neural networks and large language models, although DCNet has a small number of learnable parameters. Also, they performed simulated legion experiments and psychophysics experiments, providing intriguing similarities between biological visual systems and DCNet.

The strong influence of top-down (feedback in their study) on sensory signal processing is already well known, but this study provides some evidence that such top-down modulation can be used to build a new type of deep neural networks. The authors tested DCNet on a single task only, but this study may still contribute to advancing bio-inspired computing.

**Weaknesses:**

The paper is well written and easy to follow, but this reader finds it a little strange that the authors did not specify basic information on their model or baseline models. First, they provided mathematical descriptions of the recurrent “neuron” in the model, but it remains unclear how they implement these neurons. Based on the given source codes and the appendix, this reader thinks that individual neurons are 2D convolutional filters and that the intermediate states are used to realize recurrent behaviors. Additionally, 2D convolutional filters have multiple parameters and are not listed in the manuscript. I did find some parameters in the source codes, but they are not good enough. Second, the authors should provide more details on the baseline models. They mentioned “Traditional 6-Layer convolutional backbone feeding into a gated recurrent unit (GRU) [44] with N = 2048 neurons”, but the meaning of traditional 6-layer convolutional backbone is unclear, and they did not explain how 2048 GRU neurons provide the final answers. Once again, the authors need to provide pertinent details of each component. Third, the modulation signals are computed with two linear projects r1 and r2. It looks like they were estimated from all 4 layers, but it is unclear how they are concatenated to create r1 and r2.

**Questions:**

Please see the weakness section.

**Limitations:**

The authors provided the limitations of their study.

---

> ### Author Rebuttal · Authors · 2024-08-07
>
> We thank the reviewer for their overall positive comments. We have addressed your comments below including full model and baseline specifications. We hope that we have sufficiently addressed your concerns enough to revise your score.
>
> > **Capture the gist of interplay between low-order sensory areas and high-level cognitive areas. Top down modulation can be used to build a new class of DNNs**
>
> We sincerely thank the reviewer for this positive comment. This was indeed one of our primary motivations to pursue this line of work.
>
> > **Did not specify basic information on their model**
>
> First, we would like to thank the reviewer for checking our source code. We also acknowledge that we could have done a better job of explaining our model components. We provide a full model description in the general response (and will include this information as part of our manuscript). Here, we provide more specific details on exact model parameters.
>
> Layer 1 (Input size 3 x 128 x 128):
> Excitatory cell types (16), Inhibitory cell types (4).
> Kernel (5, 5), Padding (2, 2)
>
> Layer 2 (Input size 16 x 64 x 64):
> Excitatory cell types (32), Inhibitory cell types (8)
> Kernel (5, 5), Padding (2, 2)
>
> Layer 3 (Input size 32 x 32 x 32):
> Excitatory cell types (64), Inhibitory cell types (16)
> Kernel (5, 5), Padding (2, 2)
>
> Layer 4 (Input size 128 x 16 x 16):
> Excitatory cell types (128), Inhibitory cell types (32)
> Kernel sizes (3, 3), Padding (1, 1)
>
> Final Readout: Fully connected (1024 inputs, 6 outputs)
>
> All layers have the following convolutional kernels ($\textbf{W}$s specified in the governing equations): Input to excitation, excitation to excitation, excitation to inhibition, inhibition to excitation, and input to inhibition. The input and output dimensionalities of these convolutions are listed layer-wise above, along with the kernel and padding shapes for all convolutions in that layer.
>
> > **More details on the baseline models**
>
> Convolution filters are mentioned as: output channels x input channels x spatial size (height, width). Pooling layers are mentioned as: kernel size (height x width), stride. Inputs to every layer were normalized with LayerNorm.
>
> Layer 1 (Input size 3 x 128 x 128): Conv2d (8 x 3 x 5 x 5), ReLU, AvgPool2d (5, 5, stride 2).
>
> Layer 2 (Input size 8 x 64 x 64): Conv2d (16 x 8 x 5 x 5), ReLU, AvgPool2d (5, 5, stride 2).
>
> Layer 3 (Input size 16 x 32 x 32): Conv2d (32 x 16 x 5 x 5), ReLU, AvgPool2d (3, 3, stride 2).
>
> Layer 4 (Input size 32 x 16 x 16): Conv2d (64 x 32 x 3 x 3), ReLU, AvgPool2d (3, 3, stride 2).
>
> Layer 5 (Input size 64 x 8 x 8): Conv2d (128 x 64 x 3 x 3), ReLU, AvgPool2d (2, 2, stride 2).
>
> Layer 6 (Input size 128 x 4 x 4): Conv2d (128 x 128 x 3 x 3), ReLU, AvgPool2d (1, 1, stride 1).
>
> Project convolutional outputs to GRU: Fully connected (2048 inputs, 2048 outputs), ReLU
>
> Recurrence: GRU (2048 inputs, 2048 outputs)
>
> Final Readout: Fully connected (2048 inputs, 6 outputs)
>
> The convolutional backbone serves as a feature extractor for the GRU network. We train this baseline model by passing in the cue for the first $T$ time steps followed by the scene for the next $T$ timesteps. For our experiments we set $T=3$. A CrossEntropy Loss is used on the readout activities at the final timestep.
>
> We will include these details in the Appendix.
>
> > **Unclear modulation signal is formed from concatenating r1 and r2s**
>
> We apologize for this confusion. The modulatory signals are computed in a layer-matched manner and hence there is no issue of concatenation.
>
> However, we include in this rebuttal experimental results (in response to **qySN**) for a variant of the modulation signal that pools from all layers before computing the modulation factors. For this implementation, we linearly projected excitatory neuron activities from every layer into the dimension of interest (based on which layer is being modulated), followed by a sigmoided linear-nonlinear transformation.

---

> > ### Comment · Reviewer_BdqG · 2024-08-11
> >
> > Thank you for the clarification. This model stands between biologically realistic models, which focus on simulating realistic brain activity, and functional models, which are engineered to perform specific tasks. This type of model is rare and should be encouraged to foster meaningful interactions between neuroscience and AI research communities. As the author provided more clarification, I think it could be possible for this model to serve as a reference for those who are interested in building such models. Thus, I would like to increase my score to 6.

---

> > > ### Author Response · Authors · 2024-08-11
> > > **Thank you!**
> > >
> > > Thank you very much for your feedback and your revised evaluation recommending acceptance. In particular, thank you for highlighting the future promise of such work to serve as a bridge between neuroscientists and AI researchers.

---

### Official Review · Reviewer_qySN · 2024-07-12

**Soundness:** 3
**Presentation:** 3
**Contribution:** 2
**Rating:** 5
**Confidence:** 4

**Summary:**

The authors present a convolutional dynamical systems model if visual processing with separate excitatory and inhibitory populations in each layer. They add a low rank modulation of each layer by a factor that is computed as a linear map of the layer activities. This is meant to represent top down modulations by some higher level area. The authors train their network on visual search tasks, including a new one which requires participants to count the number of objects in a scene that match a target in color, shape, or both. The context modulation is required for good performance of the model and the model reproduces two effects from the visual search literature.

**Strengths:**

The presented model has a relatively high level of biological detail by including a split in excitatory and inhibitory pools, temporal dynamics and a low-rank top down mechanism.
Also the network is image computable, can be trained for reasonably complex tasks and the authors compare to some reaction time data.

**Weaknesses:**

All results to evaluate the network are qualitative, in contrast to claims of the authors that their network closely replicates known reaction time results.
While higher distractor heterogeneity, which makes humans slower, makes the network more uncertain (6c), target absent trials usually yield longer action times while the entropy of the network is lower here. So even the qualitative result is not as in humans.
The positive slope in 7c seems to rely heavily on the data point at 1 distractor, does this actually keep growing?
For 7d I would love some comparison to human data as the feature difference does not tell me much such that I cannot judge whether this actually matches human behaviour.
Also, It is customary to look at the accuracy of responses as well somewhere.

While the proposed network does perform better than standard DNNs the difference is small, especially to ResNet-18. Also the generalization gap is quite substantial for the proposed network as well. So overall the performance improvement is marginal.

While the authors comment on the dynamics observed in their model in Figure 5 and the accompanying text, I am not really following this part. Proper analyses of the dynamics should contain some quantitative measures of the dynamics and comparisons of these to neural data. Just looking at a set of randomly chosen activity traces is insufficient I think.

**Questions:**

Did the authors explore variants of the model where the modulation depends on all layers, has a memory or similar ideas? For a higher order representation, the current implementation seems extremely reduced.

Did the authors run any quantitative comparisons between their model and any neural or psychophysical measurements.

**Limitations:**

I agree with the authors that their work is unlikely to have negative societal impact. Limitations of their work could have been discussed more carefully though.

---

> ### Author Rebuttal · Authors · 2024-08-07
>
> We thank the reviewer for their positive comments. We believe that some of the suggestions raised by the reviewer will increase the quality of the manuscript and we now provide additional analyses and responses to these.
>
> > **Relatively high level of biological detail. The network is image computable and can be trained for reasonably complex tasks**
>
> We thank the reviewer for this note. As we noted in our response to *BT2x*, one of our goals was to tackle the increasing divergence between successful deep learning approaches that model visual computations and biological visual processing.
>
> > **Direct reaction time matches to human data**
>
> We agree with the reviewer that target-absent trials yield higher reaction times in humans while we don’t see this in our model output entropy. We wish to highlight that model entropy is far from the perfect RT metric that one can extract from neural dynamics. In fact, the subject of computing model RTs is an active research area (Spoerer et al. (2020); Goetschalckx et al. (2024); Subramanian et al. (2022); Graves (2016) to name a few), and exploration of different metrics is beyond the scope of this work. Our goal here was to expose the potential for studying model RTs in the context of cued-paradigms (through DCNet). Cued-contextual paradigms constitute a wealth of data in human psychophysical research but to our knowledge, there is no current modeling approach to study this in a naturalistic manner.
>
> As for Fig. 7c: We limit the number of distractors to 6 to account for the canvas size and placement of the objects within it. The reviewer’s point is well taken. Our response is the same as above – entropy isn’t the perfect metric. We genuinely hope to extend on this in future work.
>
> Human data comparison to 7d: While generating our stimuli, we used perceptually uniform color spaces to stay close to the experimental paradigm discussed in Wolfe and Horowitz (2004) Fig. 5a-e. We agree that this isn’t a direct comparison to human RTs and emphasize that we wished to only show human-like trends (which is strongly present in our analysis). Exact comparisons to human RTs is beyond the scope of this work, but it is something we are really interested in exploring in future work.
>
> We will add the limitations of our chosen RT metric as part of the limitations and outlook for the future in our manuscript.
>
> > **Customary to look at the accuracy of responses as well somewhere**
>
> Thanks for the comment. We do refer to the accuracy of model responses in the manuscript.
>
> L222 “DCnet achieved an overall accuracy of 97% on the test trials”.
>
> L238 “When fine tuned on this task, DCnet achieved an overall accuracy of 95% on the test trials”.
>
> > **Overall the performance improvement is marginal**
>
> As the reviewer points out, the proposed network performs marginally better than ResNet-18 in-distribution (Fig. 2b), the difference is much greater when tested out-of-distribution (Fig. 2c) – our model drops by $\sim 0.2\%$ while ResNet-18 drops by $\sim 0.6\%$. We wish to highlight that performance was not our only criteria of desire. Rather, we wanted to build a competitive and biologically-faithful framework that will allow us to study the dynamics of contextual-cueing phenomenon.
>
> > **Proper analyses of the dynamics...activity traces is insufficient.**
>
> We fully agree with the reviewer and thank them for raising this point. While comparisons to neural data is beyond the scope of this rebuttal timeline, we perform several analyses to better understand and quantify the internal dynamics of our model. We missed the point with Fig. 5, but will now update it to include the analyses we detail below (also in Fig. R2).
>
> As originally presented in Fig.5, we drive DCNet activity by uncorrelated, time-varying Gaussian noise inputs. We then compute:
>
> 1. The *Dynamic Range* (DR) of excitatory cells in each layer of DCNet and compare that to the DR of corresponding excitatory cells from DCNet (Lesioned Inhibition) (Fig. R2c). DR is computed as the Interquartile range (a measure of statistical dispersion) of a neuron’s activity over a time period of $128 ms$ when driven by noise. We take the mean over 64 trials for each neuron and plot this distribution per-layer in Fig. R2c. We find that excitatory neurons in the DCNet model have a significantly higher DR across layers compared to DCNet (Lesioned Inhibition) suggesting the role of inhibitory interactions in expanding the range of computations carried out by each neuron. A Kolmogorov–Smirnov test confirmed these significant differences (Layer 1 (statistic=0.667, p < .001); Layer 2(statistic=0.99, p < .001); Layer 3(statistic=1.0, p < .001); Layer 4(statistic=0.97, p < .001)).
>
> 1. The lag 1 autocorrelation as a measure of *stability* of the excitatory neurons (Fig. R2b). We find that DCNet excitatory neurons are significantly more stable than excitatory neurons in the DCNet (Lesioned Inhibition) model. A Kolmogorov–Smirnov test confirmed the significant difference (statistic=1.0, p < .001).
>
> 1. The E-I correlation coefficient as a measure of co-tuning in DCNet. We find that the average (across neurons) E-I correlation is as follows: -0.076 (Layer 1), 0.766 (Layer 2), 0.699 (Layer 3), 0.535 (Layer 4). This confirms the visual intuition provided in Fig. R2a that E-I co-tuning is weaker in early compared to late layers.
>
> > **Did the authors explore variants of the model...extremely reduced.**
>
> The reviewer makes an excellent suggestion. Inspired by this, we implement and test a new form of top-down modulation that pools information from all cortical layers before computing time-delayed top-down modulating factors as a non-linear transformation (one per layer) of this pooled input. This model achieved an overall accuracy of $92.35\%$ on vis-count, which is better than the metrics we report in the manuscript. However, we note that more work needs to be done to understand this better as the biological analogues of this process is not apparent.

---

> > ### Comment · Reviewer_qySN · 2024-08-08
> > **Read Rebuttal and keep rating**
> >
> > I just read through the authors responses and am going to keep the rating I gave initially, I thank the reviewers for their additional details.
> >
> > In particular like to see that the model with attention driven by all layers works better, as I think that fits our knowledge about biology better than feedback computed separately per layer.
> >
> > However, I still think proper quantitative comparisons are necessary to show that this model does indeed behave like a biological brain. If the authors believe that entropy is a bad predictor for RT, they should compare it something else like a confidence score and extract a sensible measure to predict RTs. They cite papers on how to do this. And for the dynamics I would really like to see some direct comparisons showing in what respects the dynamics are similar or different to some concrete actual measurements of brain data.
> >
> > The limited evaluation in the rating text matches my impression quite well.

---

> > > ### Author Response · Authors · 2024-08-14
> > > **further experiments with a better motivated reaction time metric**
> > >
> > > Per your comment and the current timeframe of the discussion period, we have implemented a reaction time metric inspired by evidential learning (EDL) theory, as proposed in Goetschalckx et al. (2024). Furthermore, we render a version of the task presented in Fig. 7 with up to 15 distractors (we reduce the radii of the discs to achieve this while maintaining the overall canvas size).
> > >
> > > We test our model (that we trained with EDL on the original dataset) **zero-shot** on the new stimuli to study the effect of increasing the number of distractors on the model reaction time. Our model achieves an impressive $\sim 70$% OOD generalization accuracy. We present results from "correct" trials below. We find that the linear RT trend for the low target-distractor difference trials firmly holds as we increase the number of distractors. The slope for low target-distractor trials is positive and significantly higher than that obtained for the high target-distractor trials.
> > >
> > > | # distractors     | $\xi_{cRNN}$ (Low T-D difference) | $\xi_{cRNN}$ (High T-D difference) |
> > > | ----------- | ----------- | ----------- |
> > > | 1 | 0.11 | 0.15 |
> > > | 3 | 0.21 | 0.16 |
> > > | 5 | 0.37 | 0.20 |
> > > | 7 | 0.51 | 0.21 |
> > > | 9 | 0.58 | 0.24 |
> > > | 11 | 0.69 | 0.24 |
> > > | 13 | 0.76 | 0.28 |
> > > | 15 | 0.77 | 0.31 |
> > >
> > > We hope we’ve convinced the reviewer of our framework's usefulness and general flexibility/applicability, regardless of the particular reaction time metric we choose to incorporate (a vast and ongoing area of research).
> > >
> > > We would appreciate if the reviewer could update their score if this response eases their concerns.

---

### Official Review · Reviewer_7BUE · 2024-07-14

**Soundness:** 3
**Presentation:** 3
**Contribution:** 2
**Rating:** 7
**Confidence:** 5

**Summary:**

The authors introduce DCNet, a hierarchical recurrent vision model that draws inspiration from the structure and function of biological visual circuits. This novel architecture consists of excitatory and inhibitory bottom-up neurons modulated by a (low-rank) representation of a higher area, analogous to the higher cortical and thalamic regions of the brain. Through their proposed model dynamics, the authors demonstrate emergent task-driven contextualization of bottom-up responses. Experimental results show that DCNet successfully replicates human behavioral findings on various classic psychophysics-inspired visual cue-delay search tasks, highlighting the model's potential in understanding and mimicking human visual perception.

**Strengths:**

+ The paper presents a noteworthy contribution with the proposed DCNet architecture, which creatively integrates biological anatomical insights with deep neural networks. The authors provide a clear and compelling motivation for their design, effectively addressing the need for contextual modulation of visual responses.
+ The experimental results demonstrate the promise of DCNet as a model of human visual search behavior. Notably, the findings in Figure 2 (vis-count) and Figure 6 (visual search with distractors) showcase DCNet's ability to learn generalizable contextual visual search solutions, outperforming baseline models and aligning with human behavioral patterns.
+ A particularly intriguing aspect of the paper is the emergence of repeated trajectories and attractor states for cued stimuli, as seen in Figure 3. This phenomenon offers valuable insights and warrants further exploration.
+ Figure 4 is a promising visualization validating that task-relevant information is stored in the low-rank higher area modulation. It is interesting to see how the early layer lesions are impacting shape selectivity, which one expects to emerge in higher layers of the perception stream.

**Weaknesses:**

- While the proposed DCNet model is intriguing, it would be beneficial to more explicitly highlight its anatomical constraints in relation to prior work. Conversely, the authors could further emphasize the novelty and significance of the low-rank modulation aspect, which appears to be a unique contribution. Additional signposting would help to clarify its impact on the observed results.
- To strengthen the paper, the authors may consider including ablation studies to dissect the contributions of individual components within the DCNet architecture. This would provide valuable insights into which elements are crucial for the model's high performance and alignment with human behavior.
- The authors cite relevant models in references [18-21], but it is unclear how these models perform on the vis-count task. A more comprehensive comparison would be helpful, including a detailed discussion of the differences between these models and DCNet. This would enable a more nuanced understanding of the proposed model's advantages and limitations.

**Questions:**

Please refer to my review above.

**Limitations:**

The authors have addressed limitations adequately.

---

> ### Author Rebuttal · Authors · 2024-08-07
>
> We thank the reviewer for their majorly positive comments. We believe that the suggestions raised by the reviewer will increase the quality of the manuscript and we now provide additional analyses and responses to these.
>
> > **Explicitly highlight DCNet’s anatomical constraints relative to prior work**
>
> Thanks for this comment. Also, in line with comments from the other reviewers, we now provide a full model specification in the general response. We will make sure to highlight and signpost anatomical facets of our model that are unique including: distinct E and I subpopulations, lateral recurrent feedback, learnable neuron time constants per cell-type, and long-range top-down feedback conceptualized as multiplicative low-rank perturbations.
>
> > **Ablation studies to dissect individual components of DCNet architecture**
>
> We agree with the reviewer. In this regard, we now perform two ablation studies.
>
> First, we train a version of the model where we lesion the Inhibitory cell populations in every layer. We analyze the internal input-driven dynamics of this model variant in Fig. R2. We train DCNet (Lesioned Inhibition) on vis-count (color condition; for the purposes of time). While DCNet (Lesioned Inhibition) learns the task ($95$% accurate), we find that neurons have a significantly smaller dynamic range and are comparatively less stable (details of these analyses are given in our response to **qySN**) when driven for longer time periods.
>
> Second, we train a version of the model with lesioned top-down feedback. DCNet (Lesioned Top-down) drops significantly in performance to $38$%, further highlighting the importance of top-down feedback.
>
> > **Unclear how relevant models [18 - 21] perform on vis-count task and differences between those models and DCNet**
>
> Thanks for raising this point. We did not evaluate these models on vis-count as none of these models work within a cueing-paradigm. We will, however, make sure to highlight aspects of these models that are similar to our work. [18, 19, 20] are single-layer convolutional recurrent neural network models with lateral feedback and distinct E/I populations. [21] is an extension of [20] to include a feature processing hierarchy. All these models included ML-esque normalization operations (such as BatchNorm or LayerNorm) to impart stability during training and none of these models included cell-type specific learnable integration constants as well as time-delayed top-down modulation.

---

> ### Author Response · Authors · 2024-08-12
> **feedback on our rebuttal**
>
> Thanks again for your positive evaluation of our manuscript. We hope you had a chance to review our rebuttal and the analyses presented there in. If you believe we had addressed your concerns, we would appreciate it if you can increase your score. Many thanks!

---

### Official Review · Reviewer_BT2x · 2024-07-17

**Soundness:** 2
**Presentation:** 2
**Contribution:** 2
**Rating:** 3
**Confidence:** 3

**Summary:**

The paper presents a model for how high levels of (presumably cortical) representation modulate lower levels.  A multilayer neural network with recurrent connections both within each layer and between layers is trained on a visual cue-delay-search task.  The phenomenology of the model is then studied and characterized, showing how low level representations are modulated by task demands, which makes predictions for experiments.

**Strengths:**

- Mechanisms motivated by behavioral and physiological results
- Evaluations based on psychophysical tasks
- Relatively small compared to other deep learning models
- Attempts to understand recurrent interactions in deep networks, currently missing from much of the literature.

**Weaknesses:**

- The framing of the central question of the paper in terms of a “modulatory homunculus” that must decide what features to attend to in lower-levels seems unnatural and odd.  It seems exclusively oriented toward modeling a specific laboratory task.  But what about natural vision?   It would seem these recurrent connections must always be in play for myriad tasks in daily life.  It would have been more compelling to see the model trained on a broader range of tasks - e.g., visual scene analysis - rather than this specific laboratory task.
- There are many recurrent interactions in the model, but the results focus more on characterizing the phenomenology of the model as opposed to what is learned by the recurrent weight matrices, which from a mechanistic point of view would seem most interesting.  It would also be worthwhile to start with a minimal model that demonstrates the principles of what is being learned, as opposed to throwing it all in the kitchen sink of the model shown in Figure 1.
- Because the model has so many components and it is not well-explained in the main body (including not defining all the variables in the equation - see below), by the time you get to results there is still no intuition for how the model is cued and performs the task, or how it learns.
- Unclear if the comparison to models without time dynamics and explicit cuing is informative.

Other comments:
- homunculus is misspelled both times
- 65: I don’t think there are explicit testable predictions/potential experiments that come from the model anywhere in the paper
- 110: Even if there is more model specification in appendix, one should at least define all the variables and give basic intuition for the equations. What is r_1, r_2 ?  W? Where are the time constants? Not clear how the variables correspond to interneurons, lateral connections, etc. What is the role of the spatial pooling operator (not explained even in the appendix)?
- Fig 2b: does not seem to actually outperform ResNet, even though it seems like it might have an advantage over ResNet because of cuing (though this is unclear because they don't explain cuing)
- 120: Besides the Conv RNN model (maybe?), are these comparable baselines if they have no temporal dynamics?
- 144: Don't seem to describe the novel cues and scenes anywhere. How different are they from the training data? Is a 55% accuracy impressive?
- 148: How do they know it included the cyan cylinder and not another object (and therefore conclude it's an illusory conjunction)? Could it also be confused by the occlusion of the blue cube?
- 169: "We believe"... this whole paragraph seems like speculation based on looking at Fig. 2, but no actual proof/argument?
- 184: Unclear what "cortical gradients" means, I'm assuming trajectory in PCA space? Is it that interesting/surprising that trajectories are different for color and shape given the model presumably would have to disentangle in order to get high accuracy?

**Questions:**

see above

**Limitations:**

see above

---

> ### Author Rebuttal · Authors · 2024-08-07
>
> We thank the reviewer for their feedback and insightful comments. We provide a point-by-point response below. Your comments have made the manuscript stronger. We hope that we have sufficiently addressed your concerns enough to revise your score.
>
> >**Model mechanisms are motivated by physiology/behavior and an attempt is made to understand recurrent interactions in deep neural networks which is missing from much of the literature.**
>
> We are particularly happy that the reviewer pointed this out. One of our motivations in pursuing this line of work is to tackle the increasing divergence between successful deep learning approaches that model visual computations and biological visual processing. We will make sure to highlight this in our discussion.
>
> > **What about natural vision? Why focus on a “laboratory task”?**
>
> Thanks for raising this point. It is true that feedback processes in the visual system are critical for a variety of functions. Having said that, we wish to highlight that our approach, to the best of our knowledge, is among the first to tackle contextual cueing paradigms in a realistic manner.
>
> Our focus was on the problem of how models can learn sensory representations of “cue”-ed attributes to look for in a “scene”. Please note that these attributes may not directly be present in the visual scene. For example, when cued for a specific color, that color may (or may not) exist in the scene under various lighting conditions, shadows, occlusions, object textures, material properties, and at different scales. To do this, one needs to be able to generate systematic variations on a theme. Although vis-count may not be “naturalistic” in a traditional sense, it contains non-trivial variations that are important to study. We highlight that there is a lack of similar datasets for naturalistic visual search at scale.
>
> We do hope to build towards naturalistic scale, and as a first step, we now try training our sensory backbone on naturalistic object recognition tasks (without standard ML bells and whistles such as model ensembling, test time augmentations, etc.). We report an accuracy of 84.79% on CIFAR-10 (Fig. R4). This is only a preliminary result, but ones which strongly demonstrate the potential of our framework to scale up.
>
> > **The focus is on phenomenology as opposed to gaining mechanistic insights.**
>
> Point well taken. We believe these are non-orthogonal interesting things to study. We have now performed a host of quantitative analyses to gain insights into the role of inhibitory influence in determining the dynamic range and stability of the excitatory neurons and co-tuning (please see our response to **qySN**). Additionally, we also highlight a macroscopic gradient in learned time constants in Fig. R1.
> > **``Kitchen sink” approach to Figure 1.**
>
> We wish to point out that the anatomical constraints that we include are simplified forms of well-studied elements in biophysics. Our ablation studies make it clear that the design constraints here (separate excitatory and inhibitory populations, top-down feedback, learnable dynamics, etc.) are both necessary for model performance and facilitate comparisons to relevant neuroscience literature.
>
> > **Intuition for how the model is cued and performs the task, or how it learns**
>
> We apologize for the lack of clarity in this aspect. With our revised model description (provided in the general response), we have tried to make this point clearer.
>
> Both cues and scenes are visual inputs ($\in \mathbb{R}^{128 \times 128 \times 3}$) provided to Layer 1 of our model. Cues are persistently provided to the network for the first $T$ discrete time steps followed by scenes for the next $T$ time steps. The output activities of the last layer at the last time step ($t = 2T$) is transformed into logits and a supervision signal is provided via a cross entropy loss (ground truth labels are counts from 0 to 5).
>
> Intuitively, our model learns to “up-modulate” features in the scene that resemble features of the cue, and ultimately use these up-modulated features to inform its final output. To do so, it needs to learn a disentangled feature basis (shared between cues and scenes as they are processed by the same backbone) because aspects of the same scene can be up- or down- modulated differentially based on the cue. This is what we try to expose in Figure 3.
>
> > **Unclear if the comparison to models without time dynamics and explicit cuing is informative**
>
> We believe that these comparisons are important particularly because it helps understand the upper bounds of expected performance. Our “implicit” cueing condition is one where the cue and scene are provided to the model **at that same time**. In practice, for a model without time dynamics, such as the ResNet and the Transformers, this is realized by stacking these two inputs before passing them into the model. This allows for the model to perform direct comparisons between the features of these two inputs, as opposed to time-delayed comparisons that happen in the explicit-cuing condition. Hence, this provides a potential upper-bound on training performance. Interestingly, while some of these implicit models can learn the task well (such as the ResNet in Fig. 2b), they fail to generalize (Fig. 2c).

---

> ### Author Response · Authors · 2024-08-07
> **Rebuttal, Part 2**
>
> > **homunculus is misspelled both times**
>
> Thank you for pointing this out. We have made the correction.
>
> > **No explicit testable predictions/potential experiments that come from the model anywhere in the paper (L65)**
>
> We believe our framework is among the first to link physiology to behavior (through computations) and is a necessary first step towards hypothesis generation. We will clarify this further in the manuscript.
>
> > **Lack of clarity in model specification**
>
> We apologize for this confusion. We take this comment seriously and we now spell out the entirety of our model formulation in the general response. We are happy to answer any further questions the reviewer might have.
>
> > **Fig 2b: does not seem to actually outperform ResNet, even though it seems like it might have an advantage over ResNet because of cuing (though this is unclear because they don't explain cuing)**
>
> We would like to clarify this point. In Fig. 2b what is shown is the performance of the models when the cues are in-distribution. As we point out in our answer above, the implicitly-cued models are supposed to serve as a potential performance **upper-bound**. The implicit-cue condition is computationally easier compared to the explicit-cue condition. We point the reviewer to Fig. 2c which shows that on a generalization condition, when the cues are out-of-distribution the same ResNet model falls to chance.
>
> > **Besides the Conv RNN model (maybe?), are these comparable baselines if they have no temporal dynamics?**
>
> Thanks for raising this point. As we discussed above, the baselines were primarily constructed for two reasons. First, we wanted to understand performance bounds and verify that vis-count is a non-trivial computational challenge. Second, we wanted to study the computational implications (and benefits) of two components of our framework: temporal dynamics, and top-down feedback. The Conv RNN model has temporal dynamics, but no top-down modulation. The feedforward baselines have neither.
>
> > **144: Don't seem to describe the novel cues and scenes anywhere. How different are they from the training data? Is a 55% accuracy impressive?**
>
> Thanks for raising this point. The scenes are split into a training and test set. They are i.i.d. samples from our data generation process. None of the scenes used for evaluation were seen by the model during training. We perform two types of tests with the cues. The weak generalization experiment contained cues (colors, shapes, conjunctions) that the model had seen during training (but for different scenes). The stronger generalization experiment used cues never encountered in training. Specifically, for our stronger generalization experiment, we keep the color cues green and blue; the shape cue cube; and conjunctions of these out of the training set.
>
> Chance performance for all of these tasks are $16.67$%. An accuracy of $55$% is well above chance and significantly higher than the performance of models with orders of magnitude more parameters (Fig. 2c; hatched data).
>
> We will add these clarifications to the manuscript.
>
> > **148: How do they know it included the cyan cylinder and not another object (and therefore conclude it's an illusory conjunction)? Could it also be confused by the occlusion of the blue cube?**
>
> The reviewer is correct in pointing this out. Identifying either the blue-cube or the cyan-cylinder would constitute a “binding error”. We highlighted this example since the feature dissimilarity between the target object and distractors (apart from the cyan-cylinder and blue-cube) are very high and is unlikely to be the cause of the error. However, the reviewer’s general point is well taken and we will perform an in-depth model explainability analysis. In the meantime, we will reword this sentence to reflect that this is a possibility and not a certainty.
>
> “When cued to find “blue cylinders", the model mistakenly appears to include either the cyan cylinder or the occluded blue cube in its count; which would count as an illusory conjunction”.
>
>
> > **184: Unclear what "cortical gradients" means, I'm assuming trajectory in PCA space? Is it that interesting/surprising that trajectories are different for color and shape given the model presumably would have to disentangle in order to get high accuracy?**
>
> We believe there is a misunderstanding here. In Fig. 4 we show the results of a lesion experiment we performed, where we systematically cut-out modulations from each layer of the model. Data presented in Fig. 4 are not trajectories in the PCA space. We find that lesioning later layer modulations in the model significantly impacted accuracy on the “shape” trials but not the “color” trials. Lesioning early layer modulation significantly impacted accuracy on “color” trials. From these we conclude that color selectivity emerges early while shape selectivity emerges late. The selectivity profile across layers of the model is what we refer to as a “cortical gradient”.

---

> ### Author Response · Authors · 2024-08-07
> **Rebuttal, Part 3**
>
> > **169: "We believe"... this whole paragraph seems like speculation based on looking at Fig. 2, but no actual proof/argument?**
>
> We believe that the reviewer is referring to Fig. 3 here, and not Fig. 2. In Fig. 3a, we consider model trajectories of the *same* several hundred scenes (individual trajectories shown in gray) when modulated by four different cues (subpanels). Our point is better illustrated by Fig. R3. When a given scene is cued with different colors, dynamics are driven to context-relevant states though the bottom-up responses from the scene are exactly the same. This is possible only by the inactivation of context-irrelevant subspaces.

---

> ### Author Response · Authors · 2024-08-11
> **Any response to our rebuttal?**
>
> We hope you had a chance to review our rebuttal. Given your detailed feedback and suggestions, we believe our paper has improved. Please let us know if there is anything else that we can clarify. We kindly ask you to consider revising your score if you agree that the primary concerns raised in the review were addressed.

---

### Author Rebuttal · Authors · 2024-08-07

We thank the reviewers for their time in reading our manuscript and for their extensive feedback. In this general response, we address some common themes across the reviews. We provide detailed answers to specific reviewers' comments in subsequent responses. To go with this rebuttal, we also provide a PDF with additional figures labeled **Fig. R1-R4**.

First, we sincerely thank the reviewers for their generally positive feedback. All reviewers noted the importance and novelty of our model construction, as well as the utility in understanding recurrent dynamics in deep networks and the promise of building a new class of biologically motivated models with top-down modulation.

One common critique across reviews pointed to the lack of details with respect to the model specification as well as quantitative analyses.

Towards addressing questions raised by the reviewers, we perform and include several new analyses in this rebuttal. Specifically,

1. We perform two kinds of ablation studies on DCNet. We train one version of DCNet with its inhibitory populations lesioned and one version of DCNet with top-down feedback lesioned. (**7BUE**)
1. We perform a series of mechanistic interpretability analyses to better understand the learned cell type specific time constants in our model, quantify the dynamic range and stability of excitatory neurons (in the presence and absence of inhibitory interactions), and quantify the degree of co-tuning between the excitatory and inhibitory populations in the model. (**BT2x**, **qySN**)
1. We train the DCNet backbone model on CIFAR10, an object recognition task, and report a competitive performance of $84.79$%. (**BT2x**)
1. We implement an alternative form of top-down modulation that factors in information from all layers. (**qySN**)
1. We include a detailed model specification (below) as requested by multiple reviewers (**BT2x**, **BdqG**).

We hope you agree that our manuscript has improved through your feedback and that our findings will have a significant impact on computational cognitive neuroscience.



### Model specification

Neurons in our model are either excitatory (exc) or inhibitory (inh). $i$ and $j$ are indices used to describe excitatory and inhibitory cell types respectively. $(x, y)$ describes a specific spatial location. $l$ denotes the network layer. $z^{(l)}$ is the feedforward input to layer $l$. $h$ refers to a neuron's instantaneous state. Specified below are the governing dynamics for the neurons in our model.

$$
\tau\_{\text{exc}\_i}^{(l)} \frac{d h\_{\text{exc}\_i}^{(l, x, y)}}{dt} = -h\_{\text{exc}\_i}^{(l, x, y)} + g\_{\text{exc}} ( z^{(l)}, h\_{\text{exc}}^{(l)}, h\_{\text{inh}}^{(l)}, i, x, y )
$$

$$
g\_{\text{exc}} ( z^{(l)}, h\_{\text{exc}}^{(l)}, h\_{\text{inh}}^{(l)}, i, x, y ) = f( [ W\_{\text{input} \to \text{exc}}^{(l)} z^{(l)} + \lfloor W\_{\text{exc} \to \text{exc}}^{(l)} \rfloor\_{{}\_{{}\_+}} h\_{\text{exc}}^{(l)} +
\lfloor W\_{\text{inh} \to \text{exc}}^{(l)} \rfloor\_{{}\_{{}\_-}} h\_{\text{inh}}^{(l)} ]\_{i, x, y} + b\_{\text{exc}\_i})
$$

$$
\tau\_{\text{inh}\_j}^{(l)} \frac{d h\_{\text{inh}\_j}^{(l, x, y)}}{dt} = -h\_{\text{inh}\_j}^{(l, x, y)} + g\_{\text{inh}} ( z^{(l)}, h\_{\text{exc}}^{(l)}, h\_{\text{inh}}^{(l)}, j, x, y )
$$

$$
g\_{\text{inh}} ( z^{(l)}, h\_{\text{exc}}^{(l)}, h\_{\text{inh}}^{(l)}, j, x, y ) = f( [ W\_{\text{input} \to \text{inh}}^{(l)} z^{(l)} + \lfloor W\_{\text{exc} \to \text{inh}}^{(l)} \rfloor\_{{}\_{{}\_+}} h\_{\text{exc}}^{(l)}
 ]\_{j, x, y} + b\_{\text{inh}\_j})
$$

$\tau\_{\text{exc}\_i}^{(l)}$ and $\tau\_{\text{inh}\_j}^{(l)}$ are cell-type specific learnable neural time constants. Synaptic connections $\boldsymbol{W}$s are sparse matrices on which we impose translational invariance. These are, in practice, realized as convolutions. $b\_{\text{exc}\_i}$ and $b\_{\text{inh}\_j}$ are excitatory and inhibitory cell-type specific thresholds. $f(.)$ is a non-linear activation function. We use the hyperbolic tangent as our activation function $f$. An average pooling operation $\texttt{Pool}$ is applied to layer pyramidal outputs ($h\_{\text{exc}}$) to increase the receptive field size by a factor of two.

$$
z^{(l + 1)}[t] = \texttt{Pool} ( h\_{\text{exc}}^{(l)} [t]) \odot \Gamma(\xi(h\_{\text{exc}}^{(l)}[t - T])) \\ \texttt{if}\\ t\\  \geq T \\ \texttt{else}\\  \texttt{Pool} ( h\_{\text{exc}}^{(l)} [t])
$$

We make discrete time approximations to train our model. The cue is presented first to the network ($z^{(1)}[0])$) and the dynamics are unrolled for $T$ steps followed by scene presentation ($z^{(1)}[T])$) for another $T$ steps. While the scene is presented, inputs to each layer are modulated as follows, where  $\xi(.)$ is a pooling operator that computes the average activity per cell type across all $(x,y)$. $\Gamma(\textbf{e})$ is the low-rank modulation function defined as follows:

$$
\Gamma(\textbf{e}) = \textbf{e} \odot \sigma \left( [\textbf{W}\_{l,1} \textbf{e}^{T}  + \textbf{b}\_1] \otimes [\textbf{W}\_{l,2} \textbf{e}^{T} + \textbf{b}\_2 ] \right)
$$

Here, $\textbf{W}\_{l,1}, \textbf{W}\_{l,2}$ are learnable linear projections and $\textbf{b}\_1, \textbf{b}\_2$ are learnable biases. $\otimes$ denotes outer product and $\odot$ denotes pointwise scaling. By construction, the output of $[\textbf{W}\_{l,1} \textbf{e}^{T}  + \textbf{b}\_1] \otimes [\textbf{W}\_{l,2} \textbf{e}^{T} + \textbf{b}\_2 ]$ is a low-rank matrix.

---

### Decision · Program_Chairs · 2024-09-25

**Decision:**

Accept (poster)

**Comment:**

This paper introduces a neural network framework designed to understand and implement the role of contextual modulation in visual processing. Given the extensive presence of feedback and lateral connections in the visual system and the critical role of contextual processing in both natural and artificial neural networks, I believe this work marks a significant advancement. It provides a framework for future studies exploring contextual processing through modulatory connections in both neuroscience and deep learning.

The paper received largely positive reviews. The reviewers highlighted the novel perspective and framework introduced by the authors, including the model's success in qualitatively replicating human search behavior, learning contextual search tasks, and outperforming other baseline models. However, the reviewers also raised several significant concerns regarding the study's scope and the need for additional analyses. Specifically, they pointed out issues such as the relevance of the work in naturalistic vision settings, the absence of mechanistic explanations for the model's behavior, and the lack of comparisons with similar models. From my reading of the discussions, these concerns do not appear to have been fully addressed by the authors.

While these comments are important and could certainly improve the paper, I believe that, considering the generally positive feedback from the reviewers, the paper merits acceptance as a poster presentation.